# Deciphering the Role of Wnt and Rho Signaling Pathway in iPSC-Derived ARVC Cardiomyocytes by In Silico Mathematical Modeling

**DOI:** 10.3390/ijms22042004

**Published:** 2021-02-18

**Authors:** Elvira Immacolata Parrotta, Anna Procopio, Stefania Scalise, Claudia Esposito, Giovanni Nicoletta, Gianluca Santamaria, Maria Teresa De Angelis, Tatjana Dorn, Alessandra Moretti, Karl-Ludwig Laugwitz, Francesco Montefusco, Carlo Cosentino, Giovanni Cuda

**Affiliations:** 1Department of Medical and Surgical Sciences, Magna Græcia University, 88100 Catanzaro, Italy; parrotta@unicz.it; 2Department of Clinical and Experimental Medicine, Magna Græcia University, 88100 Catanzaro, Italy; anna.procopio@unicz.it (A.P.); stefania.scalise@unicz.it (S.S.); espositoclaudia@icloud.com (C.E.); gionic90@live.it (G.N.); santamariagianluca@tum.de (G.S.); mariateresadeangelis@tum.de (M.T.D.A.); 3Klinik und Poliklinik Innere Medizin I, Klinikum Rechts der Isar, Technical University of Munich, Ismaninger Str. 22, 81675 Munich, Germany; tatjana.dorn@mytum.de (T.D.); amoretti@mytum.de (A.M.); laugwitz@mytum.de (K.-L.L.); 4DZHK (German Centre for Cardiovascular Research), Partner Site Munich Heart Alliance, 80802 Munich, Germany; 5Department of Chemistry and Pharmacy, University of Sassari, 07100 Sassari, Italy; fmontefusco@uniss.it

**Keywords:** system biology, mathematical model, arrhythmogenic right ventricular cardiomyopathy (ARVC), Wnt/β-catenin signaling, RhoA-ROCK pathway, adipogenesis, induced pluripotent stem cells (iPSCs)

## Abstract

Arrhythmogenic Right Ventricular cardiomyopathy (ARVC) is an inherited cardiac muscle disease linked to genetic deficiency in components of the desmosomes. The disease is characterized by progressive fibro-fatty replacement of the right ventricle, which acts as a substrate for arrhythmias and sudden cardiac death. The molecular mechanisms underpinning ARVC are largely unknown. Here we propose a mathematical model for investigating the molecular dynamics underlying heart remodeling and the loss of cardiac myocytes identity during ARVC. Our methodology is based on three computational models: firstly, in the context of the Wnt pathway, we examined two different competition mechanisms between β-catenin and Plakoglobin (PG) and their role in the expression of adipogenic program. Secondly, we investigated the role of RhoA-ROCK pathway in ARVC pathogenesis, and thirdly we analyzed the interplay between Wnt and RhoA-ROCK pathways in the context of the ARVC phenotype. We conclude with the following remark: both Wnt/β-catenin and RhoA-ROCK pathways must be inactive for a significant increase of *PPARγ* expression, suggesting that a crosstalk mechanism might be responsible for mediating ARVC pathogenesis.

## 1. Introduction

Heart development, function, homeostasis and remodeling are complex processes orchestrated by multiple signaling networks, some of which have been well characterized experimentally, and which interact with each other via specific, often still-unknown crosstalk mechanisms. Information about the individual pathways and how they reciprocally interact allows us to gain a complete picture of how a specific function is carried out, and which are the parts of the network that, if perturbed, e.g., through mutations, could trigger the onset of a specific pathology. In this context, mathematical modeling offers valid support for *in vitro/in vivo* experimentation, allowing, in a fast and economical way, to simulate different experimental conditions [1,2] to identify interesting hypotheses to be tested experimentally [3,4], and to reverse-engineer biological interaction networks [5,6].

Arrhythmogenic Right Ventricular Cardiomyopathy (ARVC) is an inherited heart muscle disease characterized by the progressive fibro-fatty replacement of the cardiac myocytes (CMs), leading to right ventricular failure, arrhythmia and a risk of sudden cardiac death [7]. Wnt/β-catenin is one of the best-characterized pathways involved in ARVC. However, the interaction of β-catenin with other factors such as Plakoglobin (PG) during cardiogenesis still remains not fully understood. From a genetic point of view, the majority of mutations have been observed in genes encoding for proteins of the cardiac desmosomes, i.e., desmoplakin (DSP), plakophilin-2 (PKP2), desmoglein-2 (DSG2), PG and desmocollin-2 (DSC2), which are collectively responsible for about 50% of ARVC cases. However, in a minority of patients (similarly to that reported in other forms of inherited cardiomyopathies [8]), mutations in non-desmosome genes (*DES*, *CTNNA3*, *TTN*, *TMEM43*, *LMNA*, *PLN*, *TGF*-β) have also been detected [9,10]. Desmosomes are intercellular junctions involved in the structural organization of intercalated discs, and play a critical role in protecting CMs from mechanical stress. Although they are primarily associated with cell adhesion and mechanical strength, desmosomes also mediate important signaling networks, such as the canonical Wnt pathway, whose major effector, β-catenin, is required during embryogenesis and cardiogenesis, as well as in cardiac remodeling [11,12,13]. The cytoplasmic concentration of β-catenin is finely regulated; in the absence of Wnt ligands associated with Frizzled receptors and LRP5 or LRP6 co-receptors, β-catenin undergoes constant degradation through a degradation complex consisting of Axin, APC, CK1 and GSK3, which prevents its nuclear translocation. When the Wnt pathway is active, β-catenin migrates to the nucleus, where it associates with the TCF/LEF family of transcription factors, forming a complex that binds to Wnt-responsive elements (*WREs*) on target DNA sequences [14]. The Wnt/β-catenin pathway is involved in many important cellular mechanisms, such as the control of cell proliferation, cell fate regulation and apoptosis, and it also regulates master transcription factors of adipogenesis, such as CCAAT/enhancer binding protein a (*CEBPα*) and peroxisome proliferator-activated receptor γ (*PPARγ*), which are known to be repressed upon canonical Wnt activation. Conversely, the inhibition of Wnt/β-catenin pathway initiates adipogenesis [15]. New players in canonical Wnt signaling have been identified, and among them, PG is particularly interesting due to its ability to modulate the function of β-catenin in cardiogenesis. Garcia-Gras and collaborators have demonstrated that the cardiac-specific loss of desmoplakin (DP) induces migration of the desmosome PG to the nucleus, and a two-fold reduction of the canonical Wnt signaling through TCF/LEF transcription factors [16]. PG, also known as γ-catenin, is a close relative to β-catenin and binds with TCF/LEF to form a complex interacting with DNA, though with very low affinity [17]. Based on their high sequence similarity, PG and β-catenin compete for the binding to TCF/LEF, but they have opposite effects in the regulation of the adipogenic program: the effect on gene program elicited by PG was described as mechanism responsible for ARVC, and for the lineage conversion from cardiac myocytes to adipocyte-like cells, while β-catenin binding to TCF/LEF suppresses adipogenesis. Indeed, it was reported that adult cardiac stem cells (CSCs) overexpressing PG differentiate into adipocytes, and this was postulated as a molecular event involved in ARVC pathogenesis [18]. Along with the canonical Wnt signaling, the activity of RhoA-ROCK signaling seems to play a crucial role as a negative regulator of adipogenesis through interaction with Wnt signaling, and by controlling the expression of pro- and anti-adipogenic genes [19,20]. The Rho pathway is composed of a family of GTPases that regulate the formation of actin stress fibers and modulate important cell functions, including motility, dynamic reorganization of actin cytoskeleton assembly, vesicle trafficking, progression through cell cycle, differentiation, gene transcription and apoptosis [21]. Among the Rho family members, RhoA is recognized as a molecular switch that cycles from the active GTP-bound form to the inactive GDP-bound form, and these conformations are under the control of guanine nucleotide exchange factors (GEFs) and GTPase-activating proteins (GAPs). Rho signals are passed along to its downstream effectors, among which ROCK (Rho-associated protein kinase) is the best-characterized. A crosstalk between Wnt/β-Catenin and RhoA-ROCK pathways has been postulated in ARVC [22], suggesting that these pathways might cooperate for mediating the transcriptional changes occurring in the myocardium of patients. Ellawindy and collaborators have generated a mouse model expressing a dominant-negative Rho-kinase (DN-RhoK), and their model fulfils the ARVC phenotype criteria, reporting evidence of PG migration to the nucleus of cardiac cells. Additionally, in DN-RhoK mice, two molecular players, Wnt5b and *PPARγ*, seem to be responsible for triggering adipogenesis, reinforcing the idea of a crosstalk between the two pathways [22]. The downregulation of RhoA-ROCK signaling leads to the disruption of actin stress fibers and rapid accumulation of monomeric G-actin, which binds to MLK1 (Megakaryoblastic Leukemia 1, also known as MRTFA), preventing its nuclear translocation and thus activating the expression of *PPARγ* [23]. Induced pluripotent stem cells (iPSCs) have emerged as a powerful technology for modeling human diseases in vitro [24,25,26]. Despite molecular differences between iPSCs and ESCs [27,28], a growing number of scientific reports show the utility of iPSCs in understanding the molecular basis of cardiac diseases and heart remodeling [29,30]. Using an iPSC-based strategy, the activity of RhoA-ROCK pathway was recently linked to the maintenance of an activated MRTF/SRF (Myocardin-Related Transcriptional Factors/Serum Response Factor) transcriptional program, which is responsible for the regulation of cardiomyocytes identity in a human model of ARVC [31]. Here we developed a mathematical model describing an interplay between canonical Wnt and RhoA-ROCK pathways as a molecular mechanism for triggering ARVC pathogenesis. The model takes into account mathematically: (1) the effects of competition between β-catenin and PG for the binding to TCF/LEF transcription factors on the adipogenic gene promoters, and exploring these effects by extending a previous model [32], which was modified by the addition of new reactions; (2) the analysis of the role of RhoA-ROCK signaling pathway in adipogenic gene expression; and (3) the theoretical analysis of a molecular cross-talk between Wnt/β-catenin and RhoA-ROCK pathways in ARVC. The numerical results given by the mathematical model were compared and validated with experimental data obtained using iPSC-derived CMs from a healthy individual and an ARVC patient. We conclude with three major remarks: (1) the stability of desmosomes is fundamental for the regulation of adipogenesis; (2) the simultaneous activation and proper functionality of RhoA-ROCK and canonical Wnt pathways lead to the effective inhibition of the master regulator of adipogenesis, *PPARγ*; and (3) the adipogenic phenotype cannot be induced by the overexpression of Wnt5b alone.

## 2. Results

### 2.1. Mathematical Model of Wnt/β-Catenin Pathway

The first step of our analysis aimed to interrogate the proposed model for its ability to recapitulate the experimental observations of adipogenesis suppression under MRTF/SRF (Myocardin-Related Transcriptional Factors/Serum Response Factor) the proper functionality of the canonical Wnt pathway. Thus, the initial concentration of PG and Wnt ligands were set as equal to zero (pathway in an OFF state) and two different events were considered:

1. at time = 4000 min the concentration of Wnt is set to 1 nM

2. at time = 15,000 min the concentration of Wnt is set back to zero

The stepwise activation/inactivation of the Wnt pathway was imposed in all the subsequent simulated experiments. The concentration of “adipogenic mRNA” shifts from a steady state of about 0.3 nM (active state) to 0.1 nM (inactive state). Next, we evaluated the trend of “adipogenic mRNA” in response to increasing levels of PG (from 0 to 500 nM) (Figure 1A). This simulation is based on a mechanism of double competition (competition between PG and β-catenin for TCF binding and competition between β-catenin/TCF complex and PG/TCF complex for the binding to the specific promoters) and a regulatory loop for TCF synthesis, which keeps the concentration of TCF constant. As shown in Figure 1A, increasing concentrations of PG correspond to increased levels of “adipogenic mRNA” expression both in the presence and absence of Wnt stimuli. Moreover, high concentrations of PG significantly reduce the capacity of the Wnt pathway to inhibit “adipogenic mRNA” species. In the absence of PG, there is a difference as large as 45% in the “adipogenic mRNA” expression level between the active and inactive state of Wnt pathway, while at 500 nM of PG concentration, such difference is reduced to 5%. To fully understand the mechanism of competition, we simulated a second hypothesis considering only the competition between β-catenin and PG for TCF. In this model, β-catenin/TCF complex binds to *WREs* and mediates the inhibition of adipogenesis, and thus the values relative to “adipogenic mRNA” expression results are negligibly affected by increasing levels of PG (Figure 1B). These results suggest that the competition between β-catenin and PG for TCF binding only is not sufficient to significantly affect the expression of adipogenic genes. Indeed, the competition is a double competition: β-catenin and PG compete not only for the binding to TCF, but β-catenin/TCF and PG/TCF dimers also need to compete at the transcriptional level to erase an effective adipogenic response. When regulatory feedback is present, TCF results are insensitive to increasing levels of nuclear PG. Thus, independently of the levels of PG, the concentration of the β-catenin/TCF complex in this case is sufficient to inhibit adipogenesis. To further confirm this mechanism, we generated a variant of the model in which the TCF regulatory loop is abolished, and performed simulations with the initial TCF concentration set to 15 nM. In this case, the increase of PG induces adipogenesis regardless of the presence or absence of Wnt stimuli (Figure 1C), as the increased levels of PG/TCF complexes antagonize the formation of β-catenin/TCF dimers.

### 2.2. Adipogenesis Activation Is Robust against Parameter Perturbations in the Presence of High Levels of PG

Since several numerical parameters cannot be estimated empirically, in order to have more reliable results, we performed a robust multi parametric sensitivity analysis (MPSA), allowing the identification of possible combinatorial effects of group of parameters that show a joint increase of the effects on the Adipogenic mRNA in different simulated experimental conditions (low and high concentration of PG). In particular, as shown in Figure 2, we can observe that:

At a low concentration of PG, it can be observed that a non-negligible combinatorial effect of parameters is involved in the Wnt pathway, and responsible for the maintenance of a low “Adipogenic mRNA” expression.

At a high concentration of PG, the expression of “Adipogenic mRNA” is mainly affected by parameters involved in its synthesis and degradation. In this specific case, the high initial PG concentration significantly decreases the effect of Wnt pathway to lower the expression of “Adipogenic mRNA”.

As expected, in both simulated experimental conditions, the “Adipogenic mRNA” is quite insensitive to variations of the parameters regulating the binding of PG to the degradation complex. Indeed, the inhibition of the degradation complex was shown to induce a minimal increase of PG expression with respect to β-catenin [33].

All the information concerning the Sensitivity Analysis are reported in Section 4.6 of Materials and Methods.

### 2.3. Pathways Associated with Adipogenesis Are Statistically Upregulated in ARVC

We performed a Gene Set Enrichment Analysis (GSEA) using a publicity available ARVC dataset to compare the predictions of our model with experimental data. The algorithm to perform GSEA is made available by the BROAD Institute website (http://software.broadinstitute.org/gsea/ (accessed on 10 November 2018)). The analyzed genes correspond to those belonging to the canonical adipogenic pathway (HALLMARK_ADIPOGENESIS gene set). The expression of this gene set has been statistically compared with the expression of the whole genome, and all analyses have been run with 1000 permutation. The result of GSEA (Figure 3) shows that the pathways associated with adipogenesis are statistically upregulated in ARVC, in agreement with the behavior of the model. This preliminary analysis needs to be particularized to adipogenic genes that are specifically regulated by the PG/TCF complex.

### 2.4. Biological Validation of the Trend of “Adipogenic mRNA” in PKP2^mut^ CMs during Wnt/β-Catenin Pathway Modulation

To mathematically model the Wnt/β-catenin pathway, we simulated three different hypotheses (shown in Figure 1), but only one (Figure 1A) fulfils the ARVC criteria. To biologically validate the hypothesis, CMs differentiated from PKP2^mut^ and healthy control iPSCs were treated with XAV939 and CHIR99021, Wnt pathway inhibitor and activator, respectively, for 12 days. ARVC and control CMs were then analyzed for the expression of *PPARγ* and *CEBPα,* two key activators of adipogenesis (Figure 4).

Biological validation of the in silico findings of our model was performed by considering the following condition for PG:PG = 0 for WT CMs and PG ≠ 0 for PKP2^mut^ CMs.

In PKP2^mut^ CMs, we observed a higher expression of *PPARγ* even in a steady-state condition (NT), as shown in Figure 4. When the Wnt pathway is positively modulated by treatment with CHIR99021, both adipogenic genes, *PPARγ* and *CEBPα,* resulted in downregulation compared to the untreated condition, but their expression is still higher in diseased CMs compared to the WT counterpart. Additionally, we observed that the difference in adipogenic gene expression between the untreated and treated group of CMs is higher in the control respect to PKP2^mut^ CMs. Indeed, for *PPARγ* the difference of expression between untreated and treated CMs is about 55% of NT in WT and 38% of NT in PKP2^mut^ CMs (Figure 4); for *CEBPα*, the difference of expression between untreated and treated CMs is 91% of NT in WT and 40% of NT in PKP2^mut^ (Figure 4). A significant upregulation of both adipogenic genes is observed in PKP2^mut^ CMs treated with XAV939 suggest a higher sensitivity of diseased cells to the inhibition of the canonical Wnt pathway, while the expression of adipogenic genes does not significantly change in healthy CMs. These data confirmed that, in the presence of a desmosome abnormality, the Wnt pathway capability to lower the expression of adipogenic mRNA is reduced, as predicted in the simulation described in Figure 1A.

### 2.5. Wnt/β-Catenin and Rho-ROCK Integrative Model

The complete version of our model is based on the integration of both signaling pathways (Wnt/β-catenin and RhoA-ROCK) that undergo activation/inactivation by specific enzymes and/or ligands. Leveraging the mathematical models of the two pathways, it is possible to integrate them by defining one or more putative crosstalk mechanisms, to investigate how such crosstalk affects the activation of the two pathways. As biological inputs, we have considered those species that can cause upstream activation/inactivation of the two pathways: (i) Wnt ligand activates canonical Wnt/β-catenin pathway; (ii) Wnt5b and RhoGEF activate RhoA-ROCK; and (iii) RhoGAP inactivates RhoA-ROCK. In the simulations, these inputs have been considered as constant boundary conditions; the output is represented by *PPARγ* concentration (Figure 5) as a proxy for the adipogenesis level.

The development of the integrative model was carried out through a set of simulations aimed to verify the behavior of the system in the presence of one or more stimuli. Different simulations were analyzed, and each input was set at a specific concentration. Plots relative to each simulation were grouped in two panels, one for the active state of RhoA-ROCK signaling and one for its inactive state (Figure 6A and Figure 6B, respectively). To simulate the activity of RhoA-ROCK, the concentration of Rho-GEF was set high (100 nM), while the concentration of Rho-GAP was set to 0 nM (Figure 6A). In plot 1 (orange line), the boundary conditions led to the activation of RhoA-ROCK by Rho-GEF and the activation of canonical Wnt by Wnt ligand. The expression of *PPARγ* results is low (2 nM), due to the activity of both pathways. In simulation 2 (purple line), Wnt5b cooperates with Rho-GEF to activate the RhoA-ROCK pathway, and is involved in mediating the crosstalk with Wnt/β-catenin. In spite of the inhibitory action on both pathways, the concentration of *PPARγ* slightly increases due to the action of Wnt5b, which diminishes the activity of the Wnt pathway. In plot 3 (yellow line), the input values induce the activation of RhoA-ROCK due to the action of both Rho-GEF and Wnt5b, while Wnt/β-catenin is inactive. The inhibitory effect of adipogenesis by Wnt/β-catenin is lost, and the expression of *PPARγ* increases to 11 nM. Plot 4 (blue line) simulates the condition in which RhoA-ROCK is activated by Rho-GEF and Wnt/β-catenin is inactive. The results are similar to those observed in plot 3, with little variations due to the absence of Wnt5b. Figure 6B describes the Rho-GAP-mediated inactivation of RhoA-ROCK and the activation of Wnt/β-catenin. In plot 5 (orange line) the expression of *PPARγ* is high, due to the loss of inhibition by Wnt/β-catenin in response to increased levels of PG as result of the inactivation of the RhoA-ROCK pathway. In plot 6 (purple line), the tested conditions cause inactivation of the RhoA-ROCK by Rho-GAP, activation of Wnt/β-catenin by Wnt ligands, and (re)activation of RhoA-ROCK by Wnt5b. *PPARγ* expression decreases to 4 nM, suggesting a rescue of the inhibitory effect of Wnt/β-catenin by Wnt5b-mediated activation of RhoA-ROCK. The latter induces PG to associate with the desmosome. Based on these outputs, Wnt5b alone does not appear to be able to induce adipogenesis, probably as its effect on cytoskeleton stabilization through RhoA-ROCK prevails over the competitive action of Wnt/β-catenin in response to the crosstalk reaction. In plot 7 (yellow line), both pathways, RhoA-ROCK and Wnt/β-catenin, are inactivated by Rho-GAP and Wnt ligand, respectively. The concentration of *PPARγ* is 11 nM, due to minimal inhibition by the Wnt5b-mediated activation of RhoA-ROCK. Therefore, the activation/inactivation of the RhoA-ROCK pathway through Rho-GEF and Rho-GAP, respectively, have a major effect compared to the activation mediated by the cascade of reactions involving the Wnt pathway for the planar cell polarity. In the last plot (8, blue line), the input conditions induce inactivation of RhoA-ROCK mediated by Rho-GAP and the inactivation of the WNT pathway. The concentration of *PPARγ* is very high (16 nM), due to the lack of inhibition of both pathways. From the described simulations, it results that the activation of both pathways (RhoA-ROCK and Wnt) is a mandatory condition to prevent the adipogenic differentiation of cardiomyocytes seen in ARVC patients. A ColorMap was used to group and summarize the eight tested conditions (Figure 6C).

### 2.6. Biological Validation of the Trend of “Adipogenic mRNA” in PKP2^mut^ CMs during Double Inhibition of Wnt/b-Catenin and RhoA-ROCK Pathways Modulation

To confirm the role of the Rho pathway in the pathogenesis of ARVC, we used a gene set for ARVC, including 76 human genes involved in the disease. The gene set was further processed to compute its overlapping with other available gene sets (Table 1) (e.g., Reactome, Biocarta, GO, etc.). As a major signaling pathway involved in cytoskeleton remodeling, it is plausible that an impaired functionality of the Rho pathway might be responsible for ARVC (http://software.broadinstitute.org (accessed on 10 November 2018)).

GO analysis of available gene sets shows an enrichment of signaling pathways involved in cytoskeleton remodeling, as a consequence of an impaired RhoA-ROCK pathway in ARVC pathogenesis.

The expression of *PPARγ* was analyzed after modulation of both Wnt/β-catenin and RhoA-ROCK pathways. CMs were treated for 12 days with the ROCK inhibitor, Y27632, in combination with the Wnt inhibitor, XAV939 or with the Wnt activator, CHIR99021. As predicted by our mathematical model, we observed that the condition of double inhibition (Wnt and RhoA-ROCK inhibition) led to a significant up-regulation of *PPARγ* expression, with respect to untreated cells (Figure 7, red striped bars). When cells are simultaneously treated with the ROCK inhibitor, Y27632 and CHIR99021, the level of adipogenic genes decreases in both WT and PKP2^mut^ cells. Interestingly, while in non-mutated cells the level of adipogenic genes expression returns to a level similar to that seen in their untreated counterpart, PKP2^mut^ cardiomyocytes also downregulate the expression of adipogenic genes, but to a lesser extent compared to WT cells (Figure 7, blue dotted bars). This suggests that defects in the desmosome structure hinders “adipogenic mRNA” to return to levels observed in untreated cells, strengthening the fact that a proper functionality of both pathways is mandatory for the maintenance of cardiac myocytes identity.

## 3. Discussion

Biological systems typically contain a wide range of positive and negative regulatory circuitries that have the potential to accelerate or decelerate a given biological process. In the model here developed, each input for each simulation was set to arbitrary values, allowing the prediction of the system behavior in different biological conditions. The possibility of a negative feedback was not considered in the model, due to the lack of robust evidence to support this hypothesis. However, a reduced activity of the Rho-kinase pathway was shown to contribute to ARVC development [22]. Moreover, the Rho-kinases act as intracellular regulators of Wnt signaling, suggesting that these signaling pathways may interplay and crosstalk between each other. Transcriptomic analysis of ARVC hearts provided robust evidence of dysregulation of several genes associated with Rho-signaling [34], while more recently it was demonstrated that an impaired RhoA-ROCK signaling, through the RhoA/MRTF/SRF circuit, plays a crucial role in the switch of cardiomyocytes to adipocytes [31]. Based on these experimental evidences, we developed a model for testing the variations of β-catenin in response to active/inactive states of RhoA signaling, using Rho-GEF and Rho-GAP as inputs to switch ON/OFF the pathway, and maintain the concentration of Wnt5b as equal to zero. Our model reproduces the trend of β-catenin concentrations shown in the experimental data observed in wild-type and DN-RhoK mouse hearts [22]. Different simulations were performed to profile the trend of *PPARγ* levels, allowing us to formulate a new hypothesis in association with the noncanonical Wnt ligand, Wnt5b. Since Wnt5b is used in our model as input, the resulting hypothesis can be read as a feedback mechanism. Particularly, we observed that the inhibition of RhoA-ROCK induces an increased expression of Wnt5b, suggesting that RhoA pathway activity might repress Wnt5b expression through a negative feedback, and Wnt5b in turn can activate RhoA (Figure 8A). To test this new hypothesis, we introduced an additional variable named *gWnt5b* (or mRNA encoding for the Wnt5b ligand). As for other mRNAs used in this model, for *gWnt5b* too we referred to the kinetics controlling gene regulation. The expression of *gWnt5b* is mediated by MKL1, the RhoA effector. In the absence of RhoA-ROCK activity (DN-RhoK), the steady-state concentration of Wnt5b is high, while its concentration is reduced when the pathway is inhibited (RKP down) (Figure 8B, top panel). Whether other factors cooperate with MKL1 to activate *gWnt5b* needs to be further investigated. To test the condition in which RhoA-ROCK is blocked (DN-RhoK), we have reduced two-fold the parameters relative to the reactions catalyzed by *pROK*, the phosphorylated active form of Rho kinase. Besides the lack of RhoA-kinase activity, we also tested the conditions in which the pathway is downregulated and upregulated by the activity of RhoGAP and RhoGEF, respectively. In all the analyzed cases, the concentration of Wnt was set to 1 nM. We further asked whether the kinetic described for Wnt5b could mirror the trend of *PPARγ* expression. The results clearly show that the expression of *PPARγ* increases when the RhoA-ROCK pathway is inhibited, while it decreases when the pathway is upregulated. When RhoA-ROCK is completely abolished, the expression of *PPARγ* increases dramatically (Figure 8B, bottom panel). This observation is particularly interesting, since it confirms the involvement of an impaired RhoA-ROCK pathway in the pathogenesis of ARVC. Moreover, mutations in genes encoding for proteins working downstream of the RhoA pathway, or in genes encoding the activating enzyme itself, need to be carefully investigated to delineate a clear profile of the molecular mechanisms underlying the disease. Finally, Figure 8C graphically summarizes the effects of the three simulated experimental conditions on the concentrations of the other species involved in the pathways.

## 4. Materials and Methods

### 4.1. Canonical Wnt Pathway (CWP)

Parameter identification/tuning in system biology usually involves an iterative process to develop a model that is capable of replicating a set of *in vitro/in vivo* experiments, which in turn reinforces the model’s reliability, while providing a comprehensive understanding of complex biological systems. While some parameters can be directly obtained from published experimental data, others need to be inferred and finally tested by means of experimental assays. The ARVC model herein devised uses both literature information/data and knowledge of the domain experts. Our model focuses on the link between ARVC and two well-characterized signaling pathways: the Wnt/β-catenin and RhoA-ROCK. In the first step of our model, we focused on the canonical Wnt pathway only (Figure 9), for whose development we extended a previously published model based on the prediction of β-catenin regulation by APC and Axin [32].

Reactions from 1 to 19 refer to the original model (Lee et al.). Reactions numbered from 20 to 23 in the blue box represent an extended version of Lee’s model. Reaction 20 describes the inhibition of “adipogenic mRNA” by β-catenin/TCF complex; reaction 21 refers to the regulatory loop of TCF synthesis; Reaction 22 shows the binding of PG to TCF, and Reaction 23 describes the activation of “adipogenic mRNA” by PG/TCF complex.

Figure 9 shows 19 reactions described in Lee’s model: each reaction refers to a specific biochemical event (e.g., protein binding/unbinding, activation/inactivation, transcriptional regulation). Many of the numerical values of the input quantities were experimentally extracted from Xenopus eggs data, while others were deduced from the literature, and the remaining ones were estimated on the basis of the output (degradation of β-catenin), according to the experimental data. The model developed by Lee et al. includes both ordinary differential equations and algebraic ones. Values of the parameters and initial reaction conditions and species present in the original model were adapted in this work. To extend Lee’s original model, we introduced two main mechanisms: (i) the interaction between PG and the β-catenin degradation complex; and (ii) the direct competition of PG with β-catenin for the binding to TCF. The details on the ODE system, all the species and reactions of the WCP-extended model are reported in System A and Appendix A, respectively. The mechanism of competition between PG and β-catenin is not yet completely understood, and therefore in our model we considered two different hypotheses: (1) the two proteins only compete to form a heterodimeric complex with TCF; and (2) there is also a competition of the complexes β-catenin/TCF and PG/TCF at the transcriptional level. To test the latter hypothesis, we introduced a generic “adipogenic mRNA” species, which is responsible for the adipogenic program. In basal conditions, the Wnt/β-catenin pathway is active and the complex β-catenin/TCF inhibits the transcription of adipogenic genes. Another crucial difference with respect to Lee’s model is the introduction of a regulatory feedback of TCF expression; in the original model, the concentration of TCF is assumed constant (15nM), while in our model the TCF synthesis is regulated through a positive feedback mediated by β-catenin/TCF dimer. This is an important point, since the Wnt pathway undergoes both positive and negative regulation by the induction of TCF/LEF and Axin synthesis, respectively [14]. Therefore, to investigate the effects of such positive and negative regulation, we implemented our model by introducing a “TCF encoding RNA” species, responsible for TCF production. The mathematical description of such transcriptional mechanisms has been extrapolated from another model available in literature [35]. The negative feedback introduced in our system involves the rate of Axin synthesis, whose induction depends on the availability of β-catenin and β-catenin/TCF complex [36]. In the lack of detailed biochemical information, the kinetics of the binding/unbinding of PG/TCF and the interaction between PG and β-catenin degradation complexes are assumed to be identical to those of β-catenin. The kinetics of “adipogenic mRNA” transcription was assumed to follow a classical biochemical mechanism of two transcriptional factors, competing for the same binding sites to the gene promoter (Figure 10) [37].

### 4.2. Rho Pathway (RKP)

The Rho pathway is known to be involved in adipogenesis through two distinct mechanisms: one direct, driven by the transcription factor MLK1 (Megakaryoblast Leukemia 1), and one indirect through a crosstalk with the Wnt pathway. RhoA activation does not directly affect β-catenin stabilization/accumulation, but rather acts as a “positive modifier” working synergistically with the Wnt [38]. Additionally, active RhoA enhances the activity of β-catenin by promoting its transcription activity [38,39]. We traced a set of in silico reactions, shown in Figure 11.

MLK1 shuttles between the cytoplasm and the nucleus where it transduces signals from the cytoskeleton and inhibits the expression of *PPARγ* when the RhoA-ROCK pathway is active. Reduction in RhoA-ROCK activity induces cytosolic accumulation of G-actin (globular). G-actin dimerizes with MKL1 preventing its migration to the nucleus, resulting in the repression of myogenic genes and activation of *PPARγ*.

To devise the mathematical model of Rho pathway in ARVC, we combined the two models in [40,41], and added the following function for protein phosphorylation
(1)$$v=\frac{k\cdot \;Signal\cdot \;Substrate}{K+Substrate},$$
where v is the reaction rate, *Signal* refers to the enzyme mediating the reaction and *Substrate* is the molecule that undergoes phosphorylation/dephosphorylation. The conditions used for the Wnt/β-catenin model were applied to model the activation of Dishevelled through Wnt5b and transcriptional regulation mediated by MLK1. All the information about the parameters, species and reaction of the devised model for Rho pathway are available in Appendix A [42,43], and in System B of the Appendix A, respectively. To simulate the reactions for which quantitative data were not available, we used basic mechanisms such as mass action kinetics, and specific parameters were tuned within physiological ranges. Preadipocytes exposure to a lipogenic milieu leads to the downregulation of RhoA-ROCK signaling causing a disruption of actin stress fibers and increased levels of monomeric of G-actin that interacts with MLK1, preventing its nuclear translocations, and this ultimately induces a repression of myogenic genes and activation of *PPARγ*, responsible for triggering adipogenic differentiation. Thus, reduction in the activity of RhoA-ROCK associates with increased levels of *PPARγ.* Indeed, cell lines constitutively expressing RhoA show lowered *PPARγ* expression [23]. Based on these data, we developed a Rho model for ARVC considering the following settings: (i) both RhoA-GTP and RhoA-GDP can activate ROCKs in a physiological state; and (ii) neither RhoA-GTP nor RhoA-GDP can activate ROCKs in the dominant negative state. To test different experimental evidence, we simulated a condition in which the Rho pathway is in a ON state, and the concentration of its activator was set to 100 nM, while a second simulation consisted in the abolishment of the capacity of MLK1 to inhibit the expression of *PPARγ.* Interestingly, all the conditions tested were in agreement, with the experimental data used as reference. To test the phosphorylation/dephosphorylation status, we set the following values: *K* = 100 [nmol/L] and $k=0.06\;\left[\min^{-1}\right]$, while for testing the formation of actin filaments we used the equation
(2)V=k ⋅m⋅ gActin,
where m represents a modifier, here represented by ROCK, which is responsible for the formation of actin filaments from *gActin*; $k=0.1\;\left[n{\mathrm{mol}}^{-1}\min^{-1}\right]$; a mass action with constant kinetic equal to $0.1\;\left[\min^{-1}\right]$ was used to determine the polymerization of F-actin. The results of the MPSA on the Adipogenic mRNA for RhoA-ROCK pathway are shown in Figure 12.

### 4.3. Crosstalk between Canonical Wnt and Rho-ROCK Pathways

For a comprehensive understanding of ARVC pathogenesis, we further developed a model to integrate both canonical Wnt and Rho pathways to identify a potential interplay between them. First, we evaluated the competition for binding to the same receptor, and considered the active form of Dishevelled (*Dsha*) as a unique variable. In this case, we obtained a mathematical simulation in which the canonical and the non-canonical Wnt ligands compete for *Dshi*, and induce the activation of distinct responses mediated by *cDsh* and *ncDsh* (canonical and noncanonical Dishevelled, respectively). For the crosstalk analysis relative to the desmosome stabilization by RhoA-ROCK, we have introduced a new species, *dPG* (desmosomal Plakoglobin), to distinguish it from its free form. The model was further implemented, introducing two new reactions: (1) in the presence of Rho kinase, PG associates with the desmosome; and (2) in the absence of Rho kinase, PG is free to move from cytoplasm to the nucleus and *vice versa*. Finally, based on the knowledge that Wnt5b activates the Siah2 protein (involved in ubiquitination and proteasome-mediated degradation of specific proteins), we generated a reaction in which Siah2 binds to APC, mediating the degradation of cytoplasmic β-catenin. The ODE system and all the species and reactions relative to development of the Wnt/β-catenin/RhoA-ROCK crosstalk model are shown in System c and Appendix A, respectively.

### 4.4. Cell Culture, Differentiation and Treatments

iPSCs used in this study to experimentally validate the mathematical model were generated by Professor Karl-Ludwig Laugwitz’s group (Technical University of Munich, TUM, Munich, 81675, Germany). iPSCs were generated from dermal keratinocytes from an ARVC patient who carried a heterozygous frameshift mutation (c.1760delT; p.V587Afs*655) in the *PKP2* gene encoding the desmosomal protein plakophilin, and from a healthy individual. Reprogramming of keratinocytes to iPSCs was achieved by Sendai virus-mediated transfection. Details of the iPSCs generation and characterization are reported and described in [31]. hiPSCs were cultured in mTeSR1 medium (STEMCELL Technologies, Vancouver, BC, Canada) on Matrigel-coated Petri dishes (BD Biosciences) in a humidified incubator at 37°C and 5% CO2. Cells were passed every 4–5 days with Gentle Cell Dissociation reagent (STEMCELL Technologies, Vancouver, BC, Canada). hiPSCs from both healthy control and PKP2^mut^ individual were differentiated into CMs using a PSC Cardiomyocyte Differentiation Kit (Thermo Fisher Scientific, Waltham, MA, USA), following the manufacturer’s specification. Spontaneous beating appears from day 8 to day 10 of differentiation. At day 15, CMs were dissociated into single cells using Collagenase Type II (Thermo Fisher Scientific, Waltham, MA, USA), plated onto fibronectin-coated dishes and cultured in medium composed of DMEM/F12 (Gibco), 2% FBS, 1% Glutamax Supplement (Gibco), 1% MEM Non-Essential Amino Acids Solution (Gibco), 100 µM2-β-mercaptoethanol and 0.5% penicillin/streptomycin. Thirty-days-old CMs were used for experiments. For RhoA and Wnt pathways modulation, hiPSCs-derived CMs were treated for 12 days either with XAV939 (Wnt inhibitor 10 µM, Sigma Aldrich, St. Louis, MO, USA), CHIR99021 (Wnt activator, Tocris, 3 µM) or Y27632 (Rock Inhibitor, Selleckchem, 30 µM) separately, or with a combination of Y27632 with XAV939 or Y27632 and CHIR99021.

### 4.5. Reverse Transcription PCR and Quantitative Real-Time PCR

Total RNA was extracted by TRIzol reagent (Thermo Fisher Scientific, Waltham, MA, USA) according to manufacturer’s protocol, and 1 µg was retro-transcribed using High-Capacity cDNA Reverse Transcription Kit (Applied Biosystems, Foster City, CA, USA). qRT-PCR analysis was performed with 17.7 ng of cDNA and the SensiFAST™ SYBR^®^ Hi-ROX Kit (Bioline, Meridian Bioscience, Cincinnati, OH, USA). Gene expression levels were normalized to GAPDH housekeeping gene. qRT-PCR experiment was carried onto a 384-well plate in a QuantStudio7 Pro Real-Time PCR System, and analyzed by Design and Analysis Software 2.4.3 (both from Thermo Fisher Scientific). A list of primers is provided in Appendix A.

### 4.6. Multi-Parametric Sensitivity Analysis

Most of the parameters used to model biological systems are not available in literature/databases or experimentally validated. Therefore, in order to reduce the number of parameters that need to be tuned, identified or experimentally derived, it is paramount to gain insights on the effect that such parameters exert on the model behavior.

To identify the parameters affecting the adipogenic response and that need more attention in the tuning phase, a robust MPSA was performed. In the Sensitivity Analysis, the variation of the output *yi* (referring to the concentration of “Adipogenic mRNA”) is computed with respect to a small perturbation of a specific parameter *pj*, as follows:(3)$$S_{i,j}\left(y_{i},\;p_{j},\;t\right)\;=\;\frac{\partial y_{i\;}}{\partial p_{j\;}}\frac{p_{j\;}}{y_{i}}.$$

In the MPSA, we evaluated the sensitivity of the output to simultaneous perturbation of all the parameters of the model, as follows:N = 100 random sampled parameter sets are generated from a normal distribution with a mean equal to the nominal value of each parameter, and standard deviation the 10% of this value.For each parameter set, the sensitivity of the steady-state value of “Adipogenic mRNA” with respect to each parameter is evaluated.Box-plots are used to analyze the distribution of the sensitivity for each parameter.

Moreover, in order to evaluate the sensitivity under different conditions, the MPSA was performed at both a low and high concentration of PG.

## 5. Conclusions

In this study, we propose a mathematical model to investigate the role of two key signaling pathways, Wnt/β-catenin and RhoA-ROCK, in the pathogenesis of ARVC. Our results show that nuclear accumulation of PG is accompanied by a significant increase in the expression of adipogenic master genes. Many experimental evidences suggest that a RhoA-ROCK pathway may play a critical role in ARVC pathogenesis, and recently, Dorn and co-workers have proposed the first mechanism by which mechanical signals through cell-cell contacts translates to a myocytic-to-adipocytic switch in ARVC, suggesting that cardiomyocytes identity is controlled by various pathways converging to RhoA/MRTF/SRF signaling. Based on current literature, the devised in silico model of ARVC allowed us to draw some important conclusions: (i) simultaneous activation of RhoA-ROCK and canonical Wnt pathways leads to the most effective inhibition of *PPARγ* expression, the master regulator of adipogenesis; (ii) stability of desmosomes is crucial for the regulation of adipogenesis; and (iii) the overexpression of Wnt5b alone is not sufficient to induce an adipocytic phenotype. Therefore, our study suggests that alterations of cytoskeletal filaments and/or genetic defects in the RhoA-ROCK pathway are responsible for the instability of desmosome structure, as well as for the induction of an adipogenic program in cardiac myocytes. The hypothesis of a molecular crosstalk between RhoA-ROCK and canonical Wnt pathways might represent a novel molecular mechanism underlying ARVC pathogenesis, and will drive biologists in targeting more specific and less expensive experimental procedures. Future work will be devoted to a finer assessment of the model from a mathematical point of view, including model validation/invalidation [44,45,46] through robustness analysis methods, which can be effectively used as an alternative (with respect to data fitting) means of model validation.

## Figures and Tables

**Figure 1 ijms-22-02004-f001:**
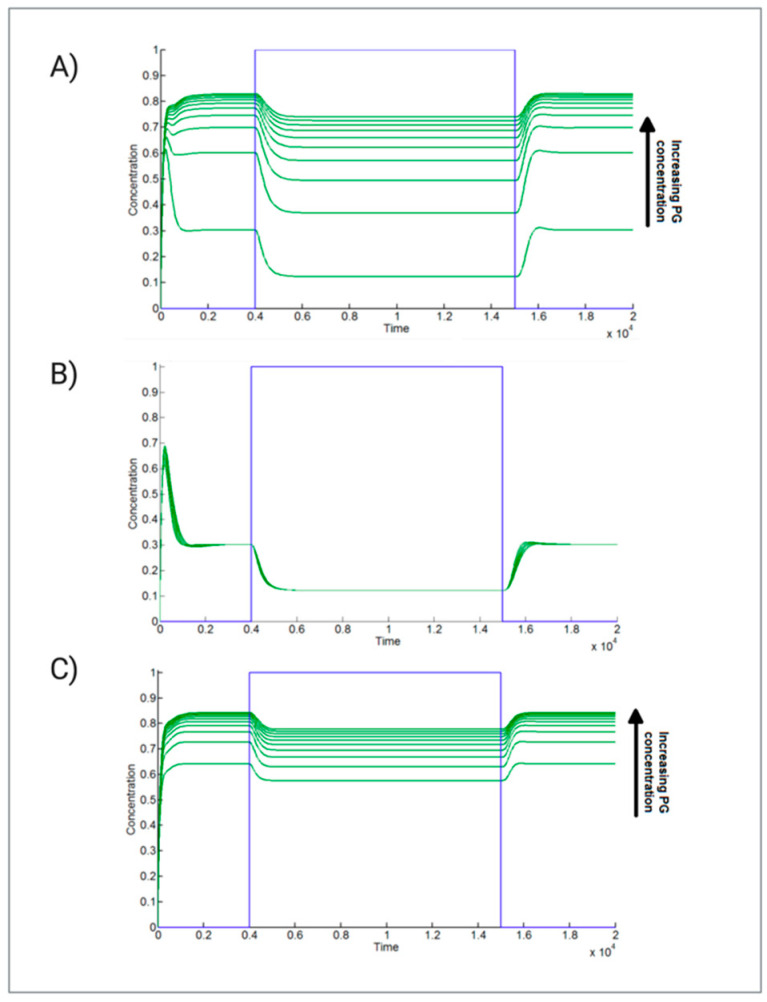
Trend of “adipogenic mRNA” expression at different concentrations of Plakoglobin (PG). (**A**) To increasing concentrations of PG (0, 56, 111, 167, 222, 167, 222, 278, 333, 389, 444, and 500 nM), it corresponds an increased level of “adipogenic mRNA”. The model takes into account a regulatory feedback for TCF synthesis and a mechanism of double competition between β-catenin and PG (they compete in both unbound and TCF-bound states). (**B**) Simulations of the model based on single competition between β-catenin and PG for the binding to TCF (no transcriptional competition between TCF-bound complexes). (**C**) Simulations of the model based on β-catenin/PG competition for TCF-binding and without TCF synthesis regulatory feedback loop (the initial concentration of free TCF is set to 15 nM).

**Figure 2 ijms-22-02004-f002:**
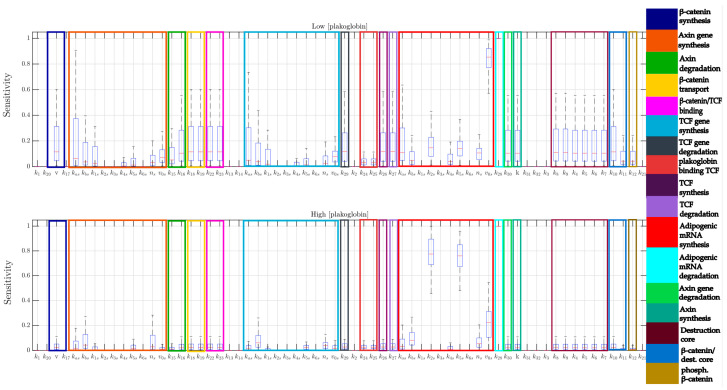
Overview of the Multi-parametric Sensitivity Analysis (MPSA) on Adipogenic mRNA for all the parameters of the WPC extended model. MPSA was performed at low (top panel) and high (bottom panel) initial concentrations of PG. The legend on the right refers to the parameters involved in each specific reaction.

**Figure 3 ijms-22-02004-f003:**
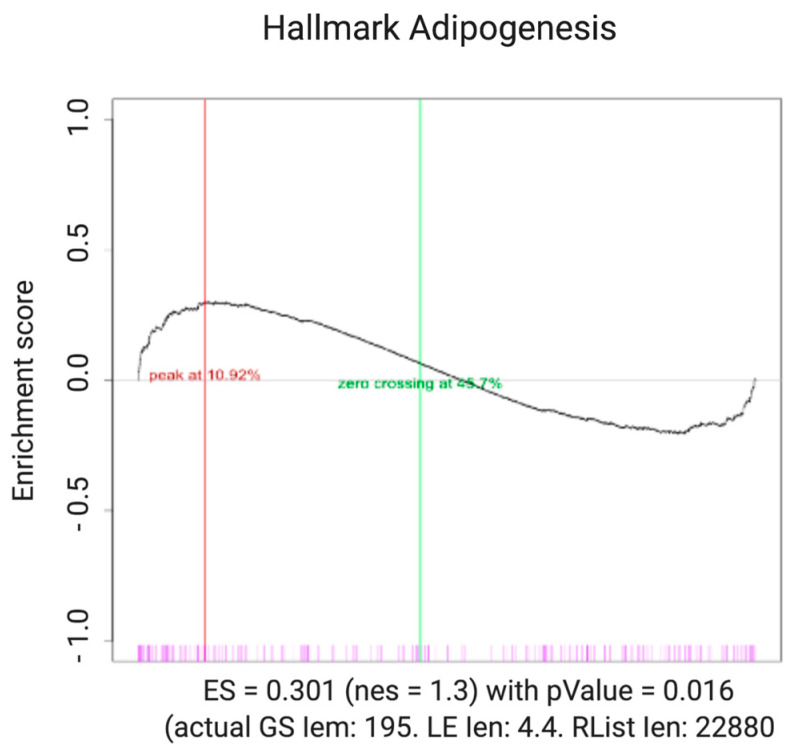
GSEA Enrichment plot (KEGG pathways) for Adipogenesis. Adipogenic signaling pathways are positively enriched in the arrhythmogenic right ventricular cardiomyopathy (ARVC) dataset for which genes belonging to the canonical adipogenic pathways (HALLMAK_ADIPOGENIC gene set) were analyzed. The black curve corresponds to the enrichment score (ES), which is the running sum of the weighted enrichment score obtained from GSEA software with the normalized enrichment score (NES) and specified *p*-value.

**Figure 4 ijms-22-02004-f004:**
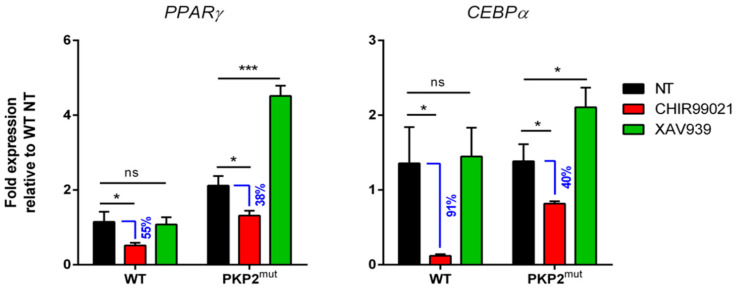
qRT–PCR analysis of *PPARγ* and *CEBPα* in PKP2^mut^ and control cardiac myocytes (CMs) cultured in presence of canonical Wnt inhibitor (XAV939) and activator (CHIR99021).WT and PKP2^mut^ CMs treated with CHIR99021 (red bars) show a downregulation of the adipogenic master regulator genes, *PPARγ* (**left panel**) and *CEBPα* (**right panel**), compared to their untreated counterpart. In agreement with the mathematical model, a higher difference of expression between untreated and treated cells was detected for the WT group and the relative percentage of expression is written in blue. XAV939 treatment (green bars) induces an upregulation of adipogenic genes in PKP2^mut^ cells only. n = 3; ns = Not significant. * *p* < 0.05,*** *p* < 0.001; *t*-test. Data are shown as means ± SEM; NT = not treated.

**Figure 5 ijms-22-02004-f005:**
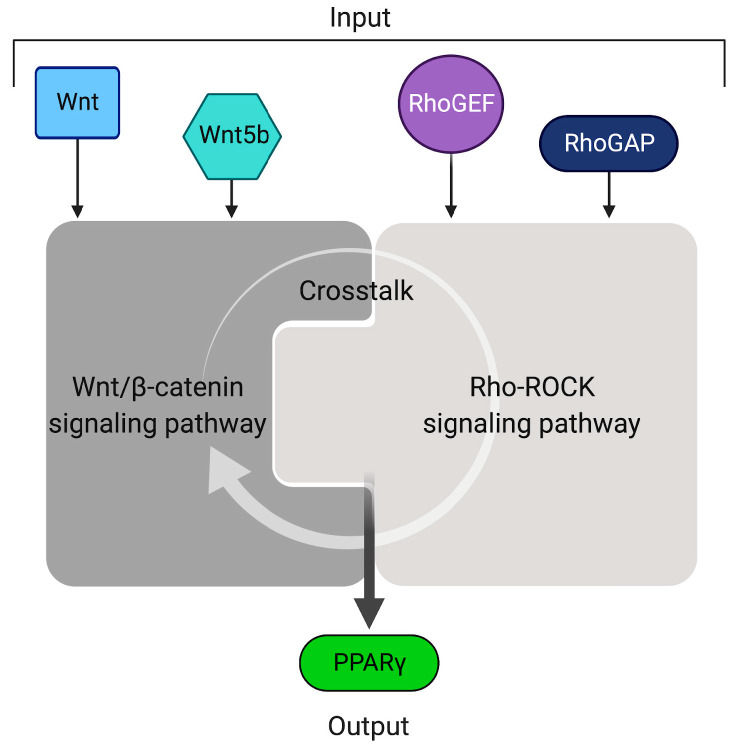
Schematic representation of integrative model for Wnt/β-catenin and RhoA-ROCK pathways. Inputs are represented by species able to activate/inactivate the two pathways (Wnt and Wnt5b for canonical and noncanonical Wnt signaling, and RhoGEF and RhoGAP for the RhoA-Rho-associated protein kinase (ROCK) signaling, respectively). Different combinations of inputs will have different effects on *PPARγ* expression.

**Figure 6 ijms-22-02004-f006:**
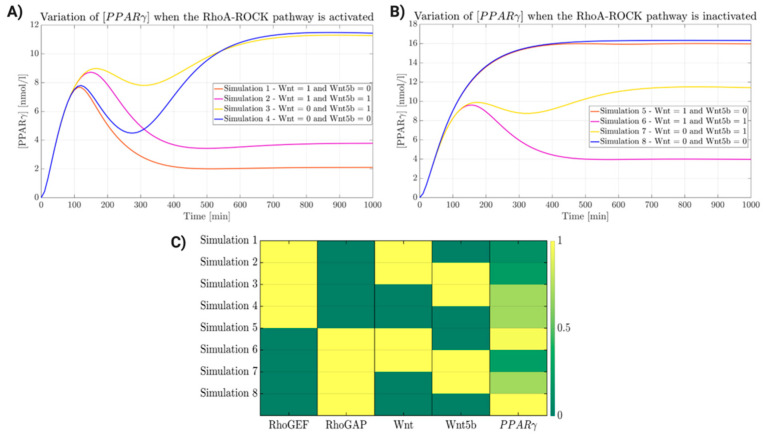
*PPARγ* variations in response to activation/inactivation of Wnt/β-catenin and RhoA-ROCK pathways. Orange, purple, yellow and blue curves are obtained by varying the concentrations of Wnt and Wnt5b. Panels (**A**,**B**) show activation (high concentration of Rho-GEF, 100 nM, and low concentration of Rho-GAP = 0) and inactivation (high concentration of Rho-GAP, 100 nM, and low concentration of Rho-GEF = 0) of RhoA-ROCK pathway, respectively. (**C**) ColorMap summarizes the effects of the eight simulated experimental conditions on the steady-state concentration of *PPAR*γ. Each column is normalized with respect to its maximum value.

**Figure 7 ijms-22-02004-f007:**
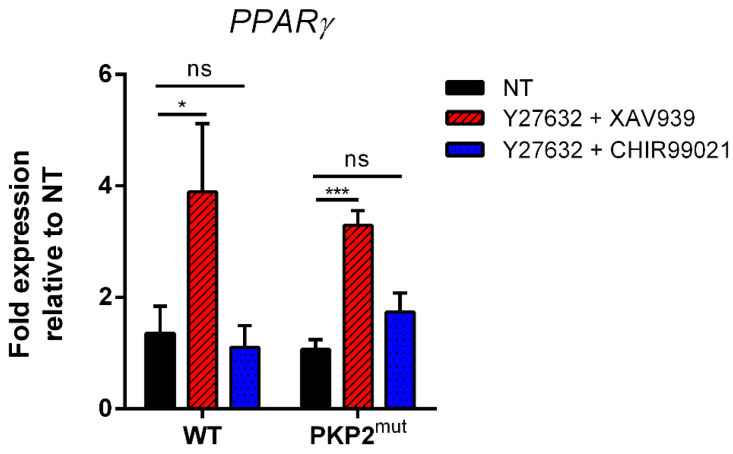
qRT–PCR analysis of *PPARγ* expression in WT and PKP2^mut^ CMs after 12 days of Wnt/β-catenin and RhoA-ROCK pathways modulation. Treatment with 30 µM Y27632 (Rock inhibitor) in combination with 10 µM XAV939 (Wnt inhibitor) leads to a significant overexpression of *PPARγ* respect to untreated CMs (red striped bars); treatment with 3 µM CHIR99021 (Wnt activator) in combination with Y27632 restored the expression of *PPARγ* to levels almost equal to untreated cells in WT group, while PKP2^mut^ CMs do not undergo to a complete *PPARγ* downregulation (blue dotted bars); n = 3; * *p* < 0.05, *** *p* < 0.001; *t*-test. Data are shown as means ± SEM; NT = not treated.

**Figure 8 ijms-22-02004-f008:**
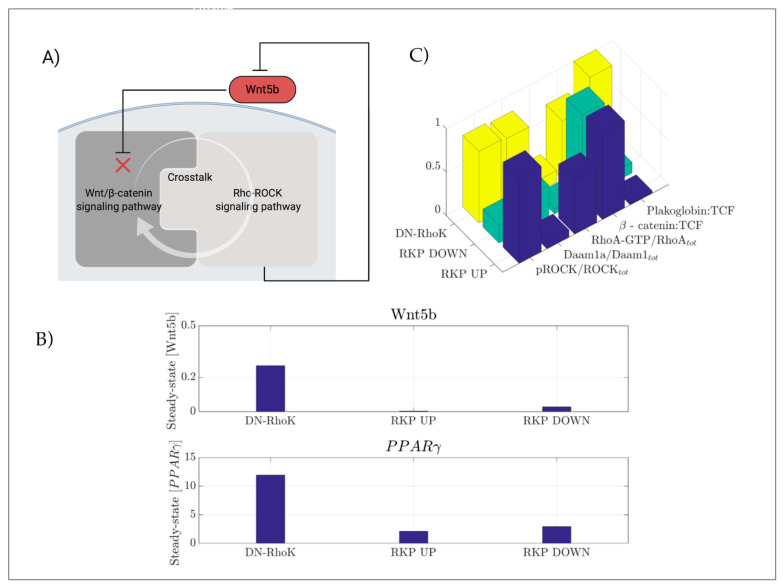
(**A**) Loop of negative feedback described in our model. (**B**) Trends of steady-state concentrations of Wnt5b (upper panel) and *PPARγ* (bottom panel) in the three conditions analyzed (DN-RhoK, RKP UP and RKP DOWN). (**C**) Concentration of pROK (Rho active kinase), active Daam1 and RhoA-GTP, with respect to the total, normalized concentrations of the β-catenin-TCF and PG-TCF complexes in the three biological situations analyzed.

**Figure 9 ijms-22-02004-f009:**
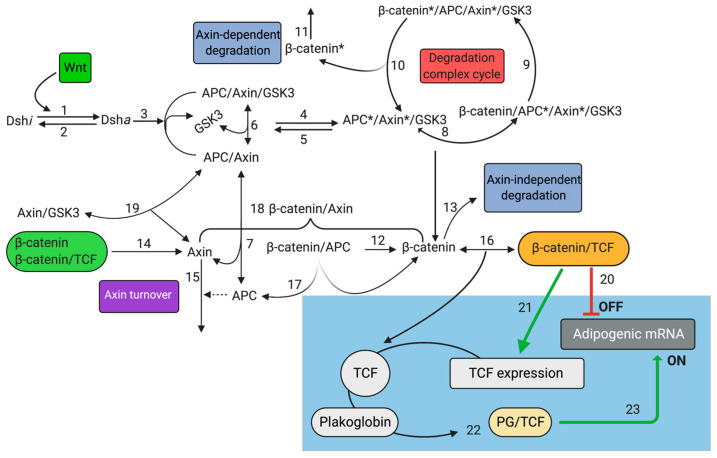
Scheme of reactions for modeling canonical Wnt signaling.

**Figure 10 ijms-22-02004-f010:**
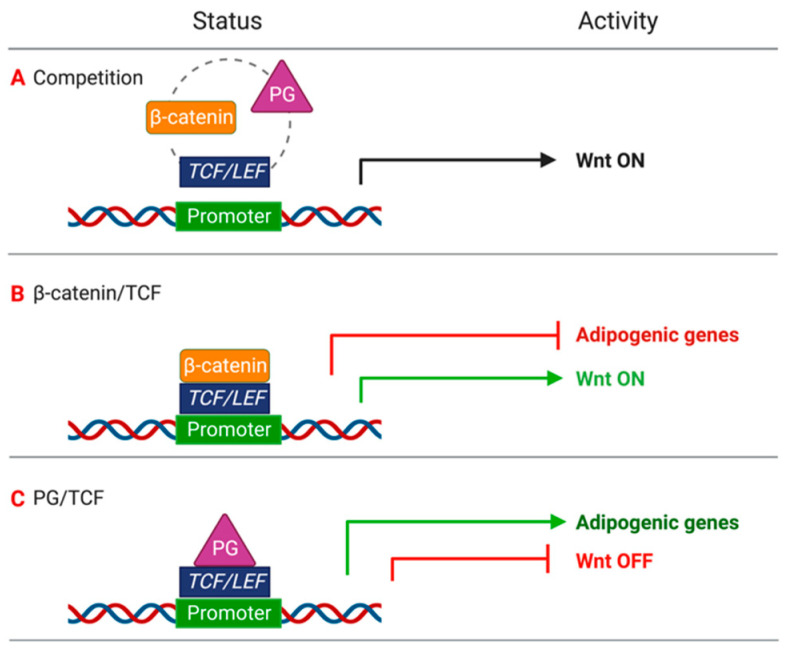
Competition of β-catenin and PG for TCF/LEF binding and relative activity. (**A**) β-catenin and PG compete for the binding to TCF/LEF; (**B**) when β-catenin/TCF binds to the promoter the transcription of adipogenic genes is inhibited; and (**C**) PG/TCF complex binds to the promoter of adipogenic genes and mediates their transcription.

**Figure 11 ijms-22-02004-f011:**
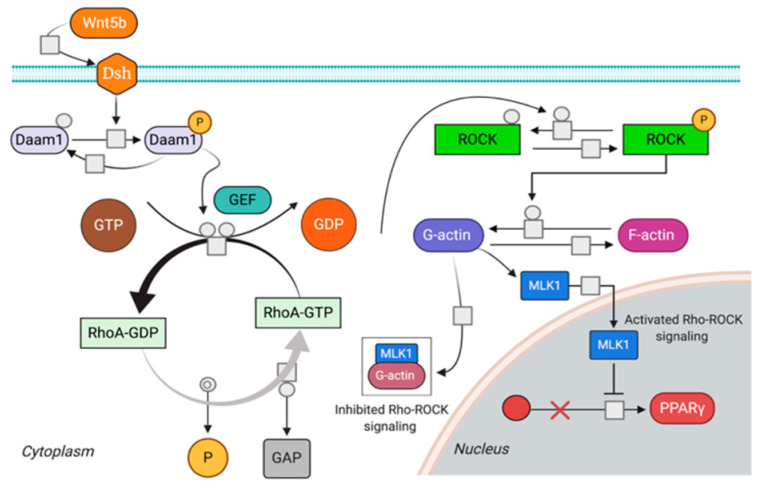
Scheme of reactions proposed for Rho pathway modelling.

**Figure 12 ijms-22-02004-f012:**
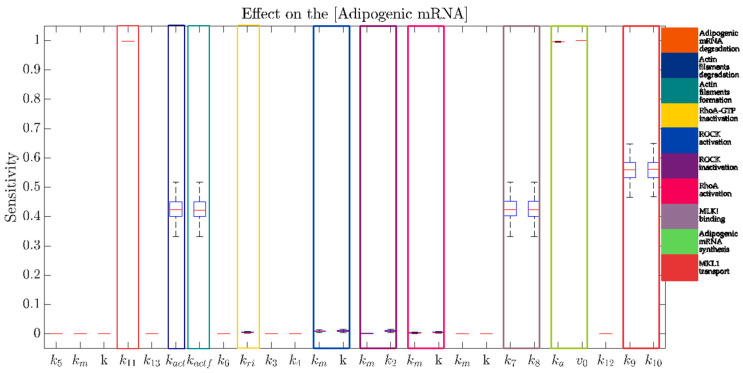
Overview of the MPSA on “Adipogenic mRNA” for all the parameters of the RhoA-ROCK model. MPSA was performed when the RhoA-ROCK pathway is activated, and shows a high sensitivity of “Adipogenic mRNA” to the parameters related to its synthesis and degradation, to the MKL1 transport and binding, and to the formation and degradation of the actin filaments. The legend on the right side describes the identified group of parameters involved in a specific reaction.

**Table 1 ijms-22-02004-t001:** Overlapping of cytoskeleton and ARVC genes.

Gene Set Name [# Genes (K)]	Description	Genes in Overlap (k)
GO_PLASMA_MEMBRANE_PROTEIN-COMPLEX [510]	Any protein complex that is part of the plasma membrane	51
GO_MEMBRANE_PROTEIN_COMPLEX [1020]	Any protein complex that is part of a membrane	53
GO_ANCHORING_JUNCTION [489]	A cell junction that mechanically attaches a cell (and its cytoskeleton) to neighboring cells or to the extracellular matrix	32
GO_CELL_JUNCTION [510]	A cellular component that forms a specialized region of connection between two or more cells or between a cell and the extracellular matrix. At a cell junction, anchoring proteins in one cell to cytoskeleton proteins in one cell to cytoskeleton proteins in neighboring cells or to proteins in the extracellular matrix	39

## Data Availability

All the data generated and/or analyzed in this study are available from the corresponding authors on reasonable request.

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
