# Peer review of "Deciphering the Role of Wnt and Rho Signaling Pathway in iPSC-Derived ARVC Cardiomyocytes by In Silico Mathematical Modeling"

_ijms, 2021, doi:10.3390/ijms22042004_

Round 1
Reviewer 1 Report
Review of the paper entitled: "Deciphering the role of Wnt and Rho signaling pathway in iPSC- derived ARVC cardiomyocytes by in silico mathematical modeling" by Parrotta et al.
In the present study, a mathematical model is proposed to investigate the role of two key signaling pathways, Wnt/β-catenin and RhoA-ROCK, and their crosstalk mechanisms in the pathogenesis of ARVC.
The study claims to have derived the following conclusions:
i) The simultaneous activation of RhoA-ROCK and canonical Wnt pathways leads to an effective inhibition of PPARγ expression, the master regulator of adipogenesis;
ii) The stability of desmosomes is crucial for the regulation of adipogenesis;
iii) The overexpression of Wnt5b alone is not sufficient to induce an adipocytic phenotype.
Consequently, it is suggested that: "alterations of cytoskeletal filaments and/or genetic defects in the RhoA-ROCK pathway are responsible for the instability of desmosome structure as well as for the induction of an adipogenic program in cardiac myocytes". The hypothesis of a molecular crosstalk between RhoA-ROCK and canonical Wnt pathways might represent a novel molecular mechanism underlying ARVC pathogenesis [...]".
**Major Concerns**
Several characteristics of the way the results are shown and explained make the evaluation of this paper in terms of the accuracy of the procedures used and general relevance hard to achieve.
In the Manuscript, several results/figures are simply validations of experimental results. A validation procedure of such type can have an informative value to decide whether or not the model can be trusted or not in their predictions when a given scenario is simulated, but by no means represents in itself a result of relevance. Consequently, if any result of relevance is present in the current Manuscript is hard to find, as it is strongly confounded by these partial "validation-results".
An additional problem in the current version of the Manuscript is that the hypotheses are not clearly stated.
The Manuscript corrections and comments were difficult to conduct, as the current version lacks line numbers, and therefore only more general points were raised.
Abstract
Unfortunately, the abstract structure is so poor that it could have been written in the absence of any result.
1 Introduction
A standard Paper structure contains a paragraph with the set of hypotheses tested and the main conclusions obtained, typically on the last paragraph. Unfortunately, the current version of the introduction lacks both.
2 Results
2.1 Mathematical model of Wnt/ β-catenin pathway
This section (and the first figure) does not show any relevant results or insight into the topic studied. As the authors explained, it merely aimed at testing if the model has the "ability to recapitulate the experimental observations of adipogenesis suppression under proper functionality of the canonical Wnt pathway.". I would suggest to move it to the Supplemental Material.
2.2 Sensitive Analysis
The title "Sensitive Analysis" would be rather suited for the Method Section. A title in the Result section should directly provide the reader with the "take-home message" of the result obtained. In this case, it should be something along the lines of: "Adipogenic mRNA is sensitive to the binding between the β-catenin and TCF and insensitive to the binding between PG and the degradation complex."
A technical note regarding the sensitive analysis:
One of the main reasons why mathematical models are essential to understand biological systems is that they allow us to carefully analyze the consequences of non-linear interaction terms. In the absence of non-linearity, intuitive verbal theories can instead be used to generate predictions. However, although non-linear interactions justify the formulation of a model, it also makes the resulting dynamics harder to understand. For instance, the change in the numerical value of a given parameter can have a consequence in the resulting dynamics if and only if the numerical value of another parameter is within a given range. In other words, in the present sensitive analysis, parameters seem to be varied one by one. This is especially the case in this model, where several numerical parameters cannot be or have not been estimated empirically. A more robust conclusion could be obtained if numerical parameters are randomized (within a realistic range), several systems are simulated, and, after a statistical analysis of the resulting system, individual as well as the interaction between parameters can be studied.
2.3 Gene Set Enrichment Analysis (GSEA)
Similar to the previous section, this one could have been entitled: "Pathways associated with adipogenesis are statistically upregulated in ARVC. ". If this result is trivial or expected, then it corresponds to a "validation type of result", better suited within the Supplemental Material.
4 Materials and Methods
Conceptually, the Materials and methods section is nicely organized, leading step by step, firstly into the Wnt Pathway, secondly into the Rho pathway, and finally into the Crosstalk between them.
The model developed in this paper is an extension of the one from Lee. Unfortunately, the information regarding the model itself, as well as the proposed extension, is fragmented into different tables. The current version forces the reader to go back and forth with the Lee manuscript, cross-check, and easily "get lost" in the nomenclature between that and this paper. It is especially unfortunate in this case, where Lee uses different nomenclature than the one used here (for variables, X's in Lee's case, and S's here). This problem can be easily overcome if the authors include the full version of the model, the 7 ODE, and the 8 algebraic equations in the Supplemental Information file. Also, in only one table, add the parameters and the initial conditions. In this way, it would be straightforward for the reader (and me as a reviewer) to reproduce the results presented here.
4.3 Crosstalk between canonical Wnt and Rho-ROCK pathways
Statements like the following:
"The model was further implemented introducing two new reactions: 1) in the presence of Rho kinase, PG associates with the desmosome; 2) in the absence of Rho kinase, PG is free to move from cytoplasm to the nucleus and vice versa."
are especially hard to track back to the system of differential equations. It would be crucial to link these conceptual statements with the structure or terms in the ODE.
Figures and Tables:
Figure 1: This figure does not show any relevant results or insight into the topic studied. As the authors explained, it merely aimed at testing if the model has the "ability to recapitulate the experimental observations of adipogenesis suppression under proper functionality of the canonical Wnt pathway.". I would suggest to move it to the Supplemental Material.
Figure 2: The Caption of this figure does not explain or even mention the main result observed.
Figure 3: This figure has merely a title, not a proper caption. How do the authors read this result?
Figure 4: Which statistical test was used?a T-test? Sample size? Degree of freedom?
Figure 5: This figure has merely a title, not a proper caption. Can the authors explain which part is newly incorporated into the proposed model?
Figure 6: It would be This figure should provide the main "Take-home message" of the paper. The current title: "Simulation of crosstalk between Wnt/β-catenin and RhoA-ROCK pathway", does not say anything about what these simulations show.
Figure 7: Which statistical test was used?a T-test? Sample size? Degree of freedom?
Figure 9: It would be helpful to add the number corresponding to the reactions into the extensions (blue box) made in this study.
Table 1: This Table is not informative at all for the main Manuscript. It can be useful in the Supplemental Material.
Table 2: The content of the caption should be the title: "Many cytoskeleton genes overlap with the ones involved in ARVC. "
Table 3: The data is presented as a barplot figure and, therefore, does not represent a Table.
Author Response
Dear Reviewer,
this is a point-by-point response made according to your suggestions. The changes we made to our previous article are highlighted using the “Track Changes” function in Microsoft Word and for each of the changes, we specified the line numbers to which it refers within the revised main text.
Confident that your comment will significantly improve the overall quality of our paper, please find below a summary of the major changes:
- the Abstract was written ex-novo providing the immediacy of purposes, prediction, and results;
- the Introduction was improved by better describing the working hypotheses, in-silico results, and experimental validation (lines 128-142).
- extended WCP (Wnt Canonical Pathway) model was reported in the Supplementary Materials. Reactions modeled from scratch are written in bold and are now clearly distinguishable from the model developed by Lee et al.
- the Sensitivity Analysis discussed in the article was strengthen and implemented by performing a robust Multi-Parametric Sensitivity Analysis (MPSA) whose results are described in Figure 2 of the revised manuscript. An explanation of the Sensitivity Analysis is provided in Paragraph 4.6;
- figures and Tables: Figures 4 and 9 were modified; legends of Figures 2, 3, 4, 5, 6, 7, and 9 were re-edited; Tables 1 and 3 were removed from the article, and Figures 2 and 12 are new.
Point-by-point revision Several characteristics of the way the results are shown and explained make the evaluation of this paper in terms of the accuracy of the procedures used and general relevance hard to achieve. In the Manuscript, several results/figures are simply validations of experimental results. A validation procedure of such type can have an informative value to decide whether or not the model can be trusted or not in their predictions when a given scenario is simulated, but by no means represents in itself a result of relevance. Consequently, if any result of relevance is present in the current Manuscript is hard to find, as it is strongly confounded by these partial "validation-results". An additional problem in the current version of the Manuscript is that the hypotheses are not clearly stated.
Response:
We agree with the reviewer’s comments that investigations and hypotheses of our study may not be clearly stated, in the first version of the manuscript. We have marked both hypotheses and final remarks in the introduction of the revised version, providing the reader with immediacy of purposes, predictions, and results.
The Manuscript corrections and comments were difficult to conduct, as the current version lacks line numbers, and therefore only more general points were raised.
Response:
We apologize for this inconvenience and fully understand that the lack of line numbers makes it difficult to correct the manuscript. Line numbers are provided in the new manuscript.
Abstract Unfortunately, the abstract structure is so poor that it could have been written in the absence of any result.
Response:
As suggested by the reviewer, the Abstract was written from scratch and provides in the current version information about the aim of the devised mathematical model and the main results obtained.
1 Introduction A standard Paper structure contains a paragraph with the set of hypotheses tested and the main conclusions obtained, typically on the last paragraph. Unfortunately, the current version of the introduction lacks both.
Response:
In accordance to the reviewer’s comment, we have modified the introduction adding, in its final part, the hypothesis and the main conclusions. Specifically, a brief step-by-step discussion of the devised mathematical model together with the experimental/biological validations was added.
2 Results
2.1 Mathematical model of Wnt/ β-catenin pathway This section (and the first figure) does not show any relevant results or insight into the topic studied. As the authors explained, it merely aimed at testing if the model has the "ability to recapitulate the experimental observations of adipogenesis suppression under proper functionality of the canonical Wnt pathway". I would suggest to move it to the Supplemental Material.
Response:
Paragraph 2.1 of the Results and Figure 1 describe a significant first result since it highlights how the new species and reactions we introduced to extend the Lee’s original model, indeed reproduce the double competition between β-catenin and PG for binding to TCF, and to the promoter of adipogenic genes, in their dimeric form. These results are new respect to the ones obtained from Lee’s model and allowed to draw a subsequent merge of the WCP model with the RhoA-ROCK one, through the identified cross-talk mechanism.
2.2 Sensitivity Analysis
The title "Sensitive Analysis" would be rather suited for the Method Section. A title in the Result section should directly provide the reader with the "take-home message" of the result obtained. In this case, it should be something along the lines of: "Adipogenic mRNA is sensitive to the binding between the β-catenin and TCF and insensitive to the binding between PG and the degradation complex”.
A technical note regarding the sensitive analysis: One of the main reasons why mathematical models are essential to understand biological systems is that they allow us to carefully analyze the consequences of non-linear interaction terms. In the absence of non-linearity, intuitive verbal theories can instead be used to generate predictions. However, although non-linear interactions justify the formulation of a model, it also makes the resulting dynamics harder to understand. For instance, the change in the numerical value of a given parameter can have a consequence in the resulting dynamics if and only if the numerical value of another parameter is within a given range. In other words, in the present sensitive analysis, parameters seem to be varied one by one. This is especially the case in this model, where several numerical parameters cannot be or have not been estimated empirically. A more robust conclusion could be obtained if numerical parameters are randomized (within a realistic range), several systems are simulated, and, after a statistical analysis of the resulting system, individual as well as the interaction between parameters can be studied.
Response:
As suggested by the reviewer, the title of paragraph 2.2 was changed as follows: “Adipogenesis activation is robust against parameter perturbations in the presence of high levels of PG”. We performed a more robust Multi-parametric Sensitivity Analysis (MPSA) to highlight the effects of simultaneous variations of the parameters. Figure 2 has been replaced by a new one containing the MPSA results. Paragraph 2.2 in the Results section, has been amended, introducing the discussion of the results obtained via MPSA (see Lines 204-226). All the methodology was moved to paragraph 4.6 of the Materials and Methods section (see Lines 756-782).
2.3 Gene Set Enrichment Analysis (GSEA) Similar to the previous section, this one could have been entitled: "Pathways associated with adipogenesis are statistically upregulated in ARVC. ". If this result is trivial or expected, then it corresponds to a "validation type of result", better suited within the Supplemental Material.
Response:
Title of subsection 2.3 was changed according to the reviewer’s suggestion, as follows: “Pathways associated with adipogenesis are statistically upregulated in ARVC.” The authors do not think the results are trivial or expected, thus they maintained the paragraph within the main text.
- Materials and Methods Conceptually, the Materials and methods section is nicely organized, leading step by step, firstly into the Wnt Pathway, secondly into the Rho pathway, and finally into the Crosstalk between them. The model developed in this paper is an extension of the one from Lee. Unfortunately, the information regarding the model itself, as well as the proposed extension, is fragmented into different tables. The current version forces the reader to go back and forth with the Lee manuscript, cross-check, and easily "get lost" in the nomenclature between that and this paper. It is especially unfortunate in this case, where Lee uses different nomenclature than the one used here (for variables, X's in Lee's case, and S's here). This problem can be easily overcome if the authors include the full version of the model, the 7 ODE, and the 8 algebraic equations in the Supplemental Information file. Also, in only one table, add the parameters and the initial conditions. In this way, it would be straightforward for the reader (and me as a reviewer) to reproduce the results presented here.
Response:
The mathematical model devised for the canonical Wnt pathway developed by Lee and co-workers was extended in our model. In particular, we have introduced two new major hypotheses: the first one concerning the competition between β-catenin and PG for their binding to TCF, the second hypothesis regards the competition between the two proteins in their dimeric form in complex with TCF (β-catenin/TCF and PG/TCF). This second competition takes place at the level of the binding of the promoters of target genes and, depending on which of the two dimeric complexes prevails in terms of abundance, the biological effect can be the induction or suppression of adipogenesis. Compared to the Lee’s model, our model includes PG as a new species and as competitor of β-catenin as well as a mechanism of regulatory feedback for TCF synthesis. We agree with the reviewer that the lack in nomenclature correspondence between the two models may cause confusion, thus we have reported the complete extended Wnt model (Please refer to System a) together with the species (Table S1) in the Supplementary Material. To make easier to track the changes to the original model, in Table S2 the reaction relative to our extension are shown in bold.
4.3 Crosstalk between canonical Wnt and Rho-ROCK pathways Statements like the following: "The model was further implemented introducing two new reactions: 1) in the presence of Rho kinase, PG associates with the desmosome; 2) in the absence of Rho kinase, PG is free to move from cytoplasm to the nucleus and vice versa" are especially hard to track back to the system of differential equations. It would be crucial to link these conceptual statements with the structure or terms in the ODE.
Response:
Desmosomal PG is introduced in the model as Dplakoglobin species (Please see Table S8 of the Supplementary Materials). The relative ODE is c.2 of the System c in the Supplementary Materials. To improve the understanding of the ODE model, we added an explanation of Dplakoglobin in the text of Supplementary Materials.
Figures and Tables: Figure 1: This figure does not show any relevant results or insight into the topic studied. As the authors explained, it merely aimed at testing if the model has the "ability to recapitulate the experimental observations of adipogenesis suppression under proper functionality of the canonical Wnt pathway". I would suggest moving it to the Supplemental Material.
Response:
Please see also our response above regarding the results of Paragraph 2.1.
Figure 2: The Caption of this figure does not explain or even mention the main result observed.
Response:
Figure 2 present in the first version of the manuscript was replaced with a new one, entitled “Overview of the Multi-parametric Sensitivity Analysis (MPSA) on “Adipogenic mRNA” for all the parameters of the WPC extended model”. A caption explaining the main results observed was added (Lines 248-251)
Figure 3: This figure has merely a title, not a proper caption. How do the authors read this result?
Response:
Title of Figure 3 was changed as follows: “GSEA Enrichment plot (KEGG pathways) for Adipogenesis” and the following caption was added: “Adipogenic signaling pathways are positively enriched in ARVC dataset for which genes belonging to the canonical adipogenic pathways (HALLMARK_ADIPOGENIC gene set) were analyzed. The black curve corresponds to the ES (enrichment score), which is the running sum of the weighted enrichment score obtained from GSEA software with the normalized enrichment score (NES) and specified p-value” (Lines 270-275).
Figure 4: Which statistical test was used?a T-test? Sample size? Degree of freedom?
Response:
We apologize for not having indicated the statistical test used. In quantitative RT-PCR analysis a t-test was used and this is now specified in the caption.
Figure 5: This figure has merely a title, not a proper caption. Can the authors explain which part is newly incorporated into the proposed model?
Response:
Figure 5 is now entitled: “Schematic representation of integrative model for Wnt/β-catenin and RhoA-ROCK pathways”. Additionally, a different caption for Figure 5 have been written: “Inputs are represented by species able to activate/inactivate the two pathways (Wnt and Wnt5b for canonical and noncanonical Wnt signaling, and RhoGEF and RhoGAP for the RhoA-ROCK signaling, respectively). Different combination of inputs will have different effects on PPARγ expression”. (Lines 366-369).
Figure 6: This figure should provide the main "Take-home message" of the paper. The current title: "Simulation of crosstalk between Wnt/β-catenin and RhoA-ROCK pathway", does not say anything about what these simulations show.
Response:
We thank the reviewer for pointing out this issue and in accordance to his comment we changed the title of Figure 6 as follows: “PPARγ variations in response to activation/inactivation of Wnt/β-catenin and RhoA-ROCK pathways”. The caption was modified as well (Please see lines 417-424).
Figure 7: Which statistical test was used?a T-test? Sample size? Degree of freedom?
Response: As for Figure 4, a t-test was used.
Figure 9: It would be helpful to add the number corresponding to the reactions into the extensions (blue box) made in this study.
Response:
We agree with this comment and thank the reviewer for his suggestion that will make easier understanding the reactions added to extend Lee’s model. Reactions shown in the blue box are now numbered (20-23) and each one is explained in the caption of the figure.
Table 1: This Table is not informative at all for the main Manuscript. It can be useful in the Supplemental Material
Response:
We agree with the fact the Table 1 is redundant thus we have deleted it form the revised version.
Table 2: The content of the caption should be the title: "Many cytoskeleton genes overlap with the ones involved in ARVC".
Response:
Table 2 in the first version is Table 1 in the new version and it is entitled: “Overlapping of cytoskeleton and ARVC genes”.
Table 3: The data is presented as a barplot figure and, therefore, does not represent a Table.
Response:
Table 3 of the first version relative to Sensitivity Analysis of the RhoA-ROCK pathway was substituted with a Figure (Figure 12) referring to the Sensitivity Analysis performed through a robust MPSA.
Reviewer 2 Report
In the present study, Parrotta and collaborators propose an in silico mathematical model to investigate the molecular mechanisms underlying arrhythmogenic right ventricular cardiomyopathy (ARVC).
ARVC is characterized by fibrofatty changes of the right ventricle, ventricular arrhythmias, and sudden death. Though ARVC is currently considered as a disease of the desmosome, desmosomal gene mutations have been identified only in half of ARVC patients, suggesting the involvement of other associated mechanisms. The authors propose two computational mathematical models to explain the activation of an adipogenic regulatory network by means of Wnt/β-catenin and Rho pathways and identified a potential interplay between RhoA-ROCK and canonical Wnt pathways as a novel molecular mechanism associated with ARVC pathogenesis. They investigated the formulated hypotheses by in silico simulations and comparison with experimental results to reveal molecular interactions that can cause the pathological state of ARVC.
These findings support the hypothesis that computational models allow us to explore the pathogenesis of complex diseases, improve our understanding of molecular mechanisms underlying disease pathology, and promote treatment strategy optimization and new drug discovery.
The manuscript is well written, the results are neat and straightforward.
I have minor comments related to the experimental validations:
- The authors claim that to biologically validate the trend of “adipogenic mRNA” during Wnt/β-catenin pathway modulation, iPSC-CMs from both PKP2mut and healthy control, were treated with XAV939 and CHIR99021, Wnt pathway inhibitor and activator. However, results in figure 4 are shown only related to Wnt activaror (CHIR99021). What is the biological effect of XAV939 in these cells?
- In figure 7 the authors show that the condition of double inhibition (Wnt and RhoA-ROCK inhibition) led to a significant up-regulation of PPARγ expression in treated compared to untreated cells, while the activation of Wnt/β-catenin pathway by CHIR99021 counteracted the effect of Y27632. However, these results look similar in both WT and mutant cells. The authors are encouraged to discuss the role of desmosome abnormality when both Wnt/β-catenin and RhoA-ROCK pathways are experimentally modulated.
Author Response
Dear Reviewer,
This is a point-by-point response made according to your suggestions. The changes we made to our previous article are highlighted using the “Track Changes” function in Microsoft Word and for each of the changes we specified the line numbers to which it refers within the revised main text.
Confident that your comments will significantly improve the overall quality of our paper, please find below our responses.
Point-by-point revision
- The authors claim that to biologically validate the trend of “adipogenic mRNA” during Wnt/β-catenin pathway modulation, iPSC-CMs from both PKP2mut and healthy control, were treated with XAV939 and CHIR99021, Wnt pathway inhibitor and activator. However, results in figure 4 are shown only related to Wnt activaror (CHIR99021). What is the biological effect of XAV939 in these cells?
Response:
We thank the reviewer for highlighting this point. Both WT and PKP2mut cardiomyocytes were treated with XAV939 and CHIR99021 for Wnt pathway inhibition and activation, respectively. As pointed out by the reviewer, Figure 4 shows the trend of expression of PPARγ and CEBP⍺ in response to Wnt activation only (CHIR99021). The explanation lies on the fact that Fig. 4 is intimately correlated with Fig. 1 which in turn considers the expression of “adipogenic mRNA” relatively to a condition of activity of Wnt pathway. As stated in the text, the expression of PPARγ and CEBP⍺ was also evaluated after treatment of CMs with XAV939 for Wnt inhibition. The revised version of our paper includes the results relative to adipogenic gene expression after XAV939 (Please see green bars in Figure 4, and lines 342-348). The expression of CEBP⍺ and PPARγsignificantly increases in the presence of XAV939 in diseased cells, while the treatment of CMs with CHIR99021 induces a downregulation of adipogenic genes in both wild-type and PKP2mut cardiomyocytes.
- In figure 7 the authors show that the condition of double inhibition (Wnt and RhoA-ROCK inhibition) led to a significant up-regulation of PPARγ expression in treated compared to untreated cells, while the activation of Wnt/β-catenin pathway by CHIR99021 counteracted the effect of Y27632. However, these results look similar in both WT and mutant cells. The authors are encouraged to discuss the role of desmosome abnormality when both Wnt/β-catenin and RhoA-ROCK pathways are experimentally modulated.
Response:
In a condition in which both pathways (canonical Wnt and RhoA-ROCK) are inactive, adipogenic gene expression increases in both analyzed groups (WT and PKP2mut cardiomyocytes). When cells are simultaneously treated with the ROCK inhibitor, Y27632, and Wnt activator, CHIR99021, the level of adipogenic genes decreases in both cell lines. Interestingly, while in wild-type cells the level of adipogenic genes expression returns to a level similar of that observed in their untreated counterpart, PKP2mut cardiomyocytes also experience a downregulation of the expression of adipogenic genes but to a lesser extend compared to WT cells. This suggests that defects in the desmosome structure hinders “adipogenic mRNA” to return to levels observed in untreated cells, strengthening the fact that a proper functionality of both pathways is mandatory for the maintenance of cardiac myocytes identity (Please see lines 452-460).
Round 2
Reviewer 1 Report
The Authors have successfully addressed the concerns raised in the previous round of revision. I do not have any comments on the present version.
A minor comment in the Supplemental Information file:
1) Page 6, 8th equation, the differential equation for Auxin, please correct "Traduzione".